# Surrogate modeling of Cellular-Potts agent-based models as a segmentation task using the U-Net neural network architecture

Tien Comlekoglu[1,2], J. Quetzalcóatl Toledo-Marín[3,4], Tina Comlekoglu[5], Douglas W. DeSimone[2], Shayn M. Peirce[1], Geoffrey Fox[6], James A. Glazier[7]*

1 Department of Biomedical Engineering, University of Virginia, Charlottesville, Virginia, United States of America, 2 Department of Cell Biology, University of Virginia, Charlottesville, Virginia, United States of America, 3 TRIUMF, Vancouver, British Columbia, Canada, 4 Perimeter Institute for Theoretical Physics, Waterloo, Ontario, Canada, 5 Beirne B. Carter Center for Immunology Research, University of Virginia, Charlottesville, Virginia, United States of America, 6 Biocomplexity Institute and the Department of Computer Science, University of Virginia, Charlottesville, Virginia, United States of America, 7 Department of Intelligent Systems Engineering, Indiana University, Bloomington, Indiana, United States of America

* glazier@iu.edu

## Abstract

The Cellular-Potts model is a powerful and ubiquitous framework for developing computational models for simulating complex multicellular biological systems. Cellular-Potts models (CPMs) are often computationally expensive due to the explicit modeling of interactions among large numbers of individual model agents and diffusive fields described by partial differential equations (PDEs). In this work, we develop a convolutional neural network (CNN) surrogate model using a U-Net architecture that accounts for periodic boundary conditions. We use this model to accelerate the evaluation of a mechanistic CPM previously used to investigate *in vitro* vasculogenesis. The surrogate model was trained to predict 100 computational steps ahead (Monte-Carlo steps, MCS), accelerating simulation evaluations by a factor of 562 times compared to single-core CPM code execution on CPU. Over short timescales of up to 3 recursive evaluations, or 300 MCS, our model captures the emergent behaviors demonstrated by the original Cellular-Potts model such as vessel sprouting, extension and anastomosis, and contraction of vascular lacunae. This approach demonstrates the potential for deep learning to serve as a step toward efficient surrogate models for CPM simulations, enabling faster evaluation of computationally expensive CPM simulations of biological processes.

**Data availability statement:** All code used to generate data and to run experiments and for model training and figure generation are available in a GitHub repository at https://github.com/tc2fh/CPM_UNet_Surrogate. We have also used Zenodo to assign a DOI to the repository: https://zenodo.org/records/15399533.

**Funding:** JAG received salary support from National Science Foundation grants 2303695 and 2120200 to Indiana University https://www.nsf.gov/. DWD received salary support from National Institutes of Health grant GM131865 to the University of Virginia https://www.nih.gov/. TC received salary support from National Institutes of Health grants T32-GM145443 and T32-GM007267 to the University of Virginia https://www.nih.gov/. GF received salary support from Department of Energy grant ASCR DE-SC0023452 to the University of Virginia https://www.energy.gov/science/ascr/advanced-scientific-computing-research. The funders had no role in study design, data collection and analysis, decision to publish, or preparation of the manuscript.

**Competing interests:** The authors have declared that no competing interests exist.

## Author summary

The Cellular-Potts model is a powerful and ubiquitous framework for developing computational models for simulating complex multicellular biological systems. Cellular-Potts models (CPMs) are often computationally expensive due to the explicit modeling of interactions among large numbers of model agents as well as diffusive fields described by partial differential equations (PDEs). In this work, we develop a convolutional neural network (CNN) surrogate model using a U-Net architecture that accounts for periodic boundary conditions. We use this surrogate to accelerate the evaluation of a mechanistic CPM previously used to investigate in vitro vasculogenesis by a factor of 562 times compared to single-core CPM code execution on CPU. For up to three recursive evaluations, or 300MCS, our model captures the emergent behaviors demonstrated by the original mechanistic Cellular-Potts model of vessel sprouting, extension and anastomosis, and contraction of vascular lacunae. Our approach demonstrates a step towards the development of surrogate models for CPM simulations using deep neural networks, enabling faster evaluation of computationally expensive CPM simulations of biological processes.

## Introduction

Multicellular agent-based models are commonly used in systems biology to investigate complex biological phenomena. These models often require calculating behaviors of many model objects or agents at once. Each agent often represents individual cells, where each cell responds to other cells in its environment. Simulating large and complex biological phenomena with many cells at once results in models that are computationally expensive. Significant computational expense results in models that are more difficult for a user to investigate or continue to develop upon.

The Cellular-Potts method is one such computational modeling method that has allowed for the agent-based simulation and *in silico* investigation of complex biological processes. Recent works with the Cellular-Potts modeling method has allowed for the *in silico* study of the mechanisms behind many complex biological processes such as the growth of blood vessels [1], the regeneration of injured muscle [2], and the development of the embryo [3–5]. In many of these works, *in silico* cell agents respond to each other as well as to diffusive fields described by systems of partial differential equations (PDEs), which further increases computational demand. The Cellular-Potts method models cell motility using a stochastic modified metropolis monte-carlo algorithm. This algorithm has been cited to be computationally expensive to perform for biological simulations involving many cells at large spatial scales [6–8]. Coupling PDEs for modeling biological systems additionally adds to the computational complexity of these *in silico* models. While effective at representing complex biological systems, the capabilities of the Cellular-Potts modeling method have also resulted in slow to evaluate mechanistic computational models. Efficient

surrogate models to approximate and accelerate such modeling methods would facilitate the accurate simulation of these biological systems at larger spatial scales or longer time scales.

Deep neural-network based surrogate models may provide an effective approach to accelerating computational model evaluation of Cellular-Potts models. Deep learning models have already demonstrated potential as effective surrogates to solve systems of PDEs for physical systems such as heat transfer and molecular and subatomic particle dynamics [9–13]. Additionally, previous work by the authors demonstrated the potential for deep convolutional neural networks to effectively solve for the steady state diffusion governed by a system of PDEs [14,15]. However, the development of neural network surrogates for Cellular-Potts agent-based models have not yet been thoroughly investigated.

Developing a neural network surrogate for the Cellular-Potts method has not yet been accomplished. The Cellular-Potts method results in stochastic agent-based models, whereas previous neural-network based surrogates, such as those mentioned for solving PDEs, use deterministic methods. Additionally, agent-based models are often used to investigate emergent behaviors which are not explicitly described or encoded in the model. In this work, we build upon our former efforts to apply convolutional neural networks as model surrogates for a steady state diffusion solver. Here, we apply our U-Net to predict the agent-based mechanistic model configuration 100 computational timesteps ahead of a stochastic CPM model formerly used to investigate vasculogenesis, where the behaviors of the CPM model agents react to diffusing cytokines described by systems of PDEs [1]. Our work demonstrates the efficacy of predicting the time-evolution of a stochastic agent-based model using a deterministic neural network architecture.

## Results

### Problem definition and approach

The selected Cellular-Potts model representing vasculogenesis demonstrates consistent patterns over the course of the simulation. These patterns are 1) sprouting of new vessel growths, 2) extension of vessel sprouts into large lacunae that anastamose with other sprouts, resulting in subdivision of large lacunae into smaller lacunae, and 3) shrinking of smaller lacunae as previously described in Merks et al. [1]. At short timescales, on the order of single computational timesteps (Monte Carlo Steps, MCS), these patterns are obscured by stochastic noise inherent to the Potts algorithm, making it challenging for a surrogate model to predict. However, at long timescales, the initial or reference configuration may be lost due to large positional changes in the vascular network during simulation. This similarly yields a challenging problem for a surrogate model to predict.

We instead attempt to capture these three model behaviors by learning an intermediate timescale with our model surrogate. We found that at 100 MCS increments, the vascular network deviated by approximately one Cellular-Potts agent length. This intermediate timescale presents a feasible problem for the surrogate model to learn. In the intermediate configuration, the system has not deviated so much from the reference configuration that the reference position is no longer apparent. The intermediate timescale also captures the three behaviors of sprouting of new vessel branches, extending and merging of existing vessel branches, and closing of smaller vascular lacunae described by the Cellular-Potts model as demonstrated in Fig 1. These three behaviors occur consistently and predictably at the 100 MCS timescale and are therefore predictable enough that we may expect our model to be successful at learning these patterns. Furthermore, deterministic PDE models [25–27] have previously been used to model this biological system, thus we would expect our U-Net model to be able to learn these behaviors.

We framed the prediction of a future Cellular Potts model state as a segmentation problem, treating the reference configuration and future simulation state similarly to predicting a labeled mask from a natural image. Convolutional neural networks, and especially the U-Net architecture, have been demonstrated to be well suited to these segmentation tasks [16,17].

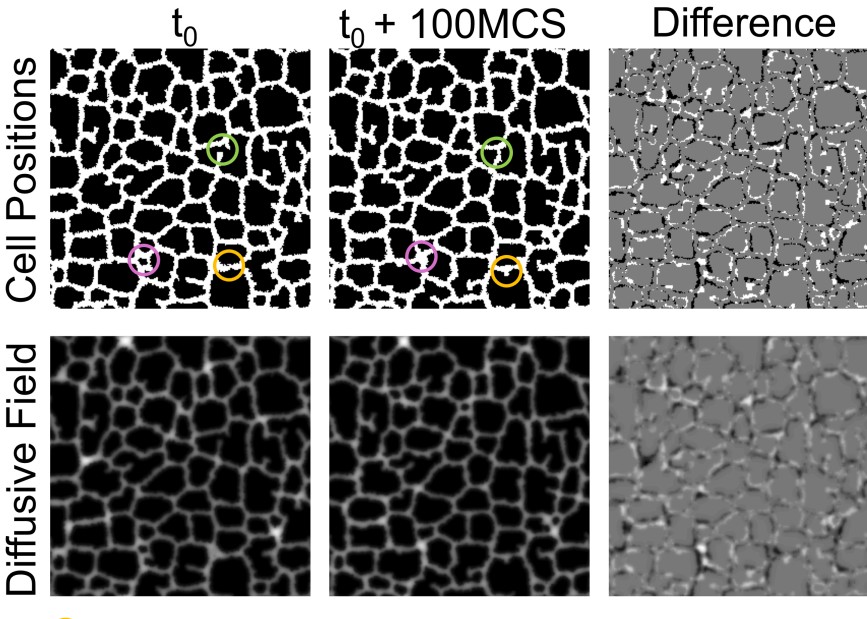

| $t_0$ | $t_0 + 100MCS$ | Difference |

○ - Sprouting of new vessel branches
○ - Extension and merging of branches
○ - Shrinking/closing of small lacunae

**Fig 1**. **Problem definition for the model surrogate.** Cell positions (top row) and diffusive field concentrations (bottom row) from an example reference configuration ($t_0$, left) and its corresponding configuration 100 MCS ahead ($t_0$ + 100 MCS, middle). The difference between the two configurations is shown (right) demonstrating a positional movement of approximately Cellular-Potts model cell length of the vascular network. This representative example demonstrates three main behaviors of the Cellular-Potts model of 1) sprouting new vessel branches, 2) extension and anastomosis of vessel sprouts, and 3) closing of small lacunae.

### Surrogate predictive performance evaluation

A given Cellular-Potts model simulation state consists of both a layer containing information about the position of the vessel network and a layer containing information about diffusive field concentrations that the vessel network responds to. After U-Net specification and training as described in Methods, we applied the U-Net recursively for multiple iterations to evaluate surrogate model performance visually. A representative diagram of recursive evaluation of a given simulation state is displayed in Fig 2. Movies 1 and 2 demonstrate extending this recursive evaluation for 98 iterations to predict simulation states in increments of 100 up to MCS 11800 using a reference configuration of MCS 2000 from a simulation not included in the training data. S1 Movie displays the results of the surrogate model prediction of the vessel network configuration, and S2 Movie displays the prediction of the chemical field concentration.

To quantitatively evaluate the surrogate predictions of the position of the vascular network for multiple simulations, we used the Sorenson-Dice coefficient (Dice score) which is a metric commonly used for evaluating the accuracy of segmentation tasks [18–21]. We used mean square error (MSE) to evaluate the accuracy of the predictions of the diffusive field component. Additionally, we calculated differences in the distributions of lacunae area of the model prediction and the reference configuration using Wasserstein's distance, also known as Earth Mover's Distance, EMD. The Earth Mover's Distance is a common metric used to quantify the distance between distributions of data, such as similarity of images, for machine learning and computer vision applications [22–24]. Metrics calculated from the surrogate model predictions versus the ground truth simulation state at the predicted MCS were compared with the same metrics comparing the reference configuration to the ground truth simulation state at that MCS. We use this comparison to determine whether the

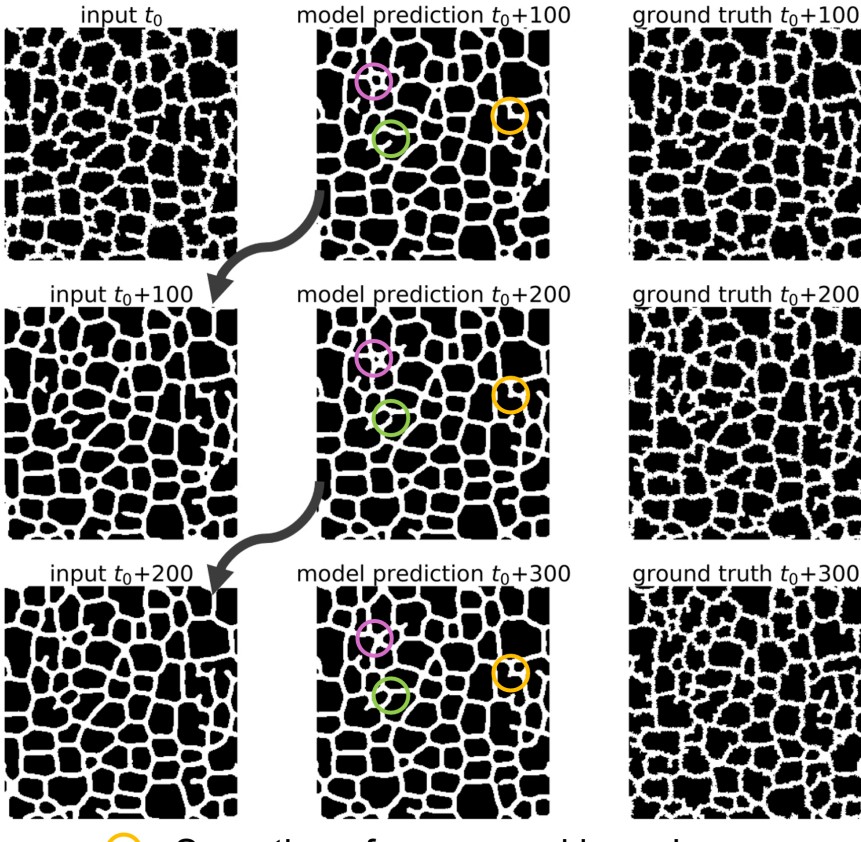

- 🟠 - Sprouting of new vessel branches
- 🟢 - Extension and merging of branches
- 🟣 - Shrinking/closing of small lacunae

**Fig 2. Recursive evaluation of a trained surrogate model.** The trained surrogate model was applied iteratively to predict hundreds of MCS ahead of a reference configuration can capture underlying Cellular-Potts model dynamics. Reference input configuration $t_0$ is from a reference simulation state at MCS 2000 from a representative Cellular-Potts model simulation generated separately from the data used for training the surrogate.

surrogate outperforms the unchanged reference configuration based on quantitative metrics. If the surrogate model outperforms the unchanged reference, it would suggest that the surrogate has learned patterns that are at least minimally predictive towards the Cellular-Potts model's true future state.

To evaluate the performance of the surrogate model, we provided an initial configuration at MCS 2000 to the surrogate model to predict forward from and performed recursive predictions for 100 iterations. This resulted in predictions for up to 10000 MCS ahead at increments of 100 MCS. Metrics of comparison to quantify the ability for the surrogate to predict the Cellular-Potts model configuration are shown in Fig 3. Data was generated using MCS 2000 from 25 unique simulations generated for surrogate evaluation. Mean and standard deviation for the data are displayed in Fig 3.

When evaluated recursively on the evaluation dataset, the surrogate model predictions resulted in a mean Dice score of 0.82, (s.d. $4.496 \times 10^{-3}$, n=25) versus 0.77 (s.d. $5.83 \times 10^{-3}$, n=25) of the input reference configuration as compared to ground truth for a single prediction step of 100 MCS. For this single prediction step, diffusive field MSE were 0.021 (s.d. $1.6 \times 10^{-3}$, n=25) compared to a MSE of of 0.038 (s.d. $4.22 \times 10^{-3}$, n=25). Quantified distances in the distribution of the lacunae areas for predicted, reference, and ground truth simulation configurations resulted in mean EMD values of 20.41

PLOS Computational Biology

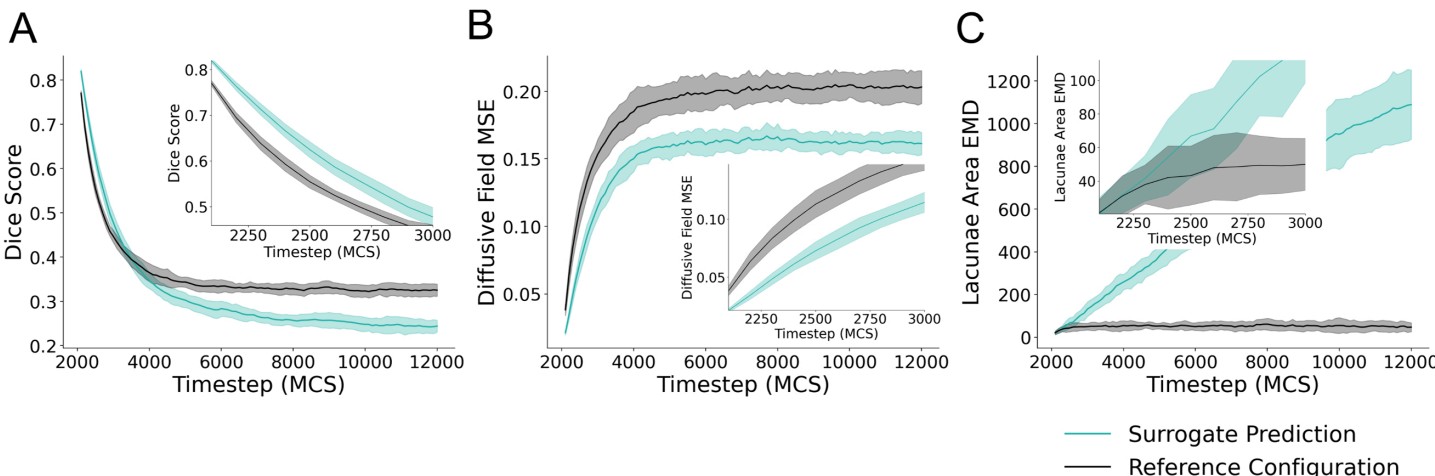

**Fig 3**. **Quantitative evaluation of the surrogate model.** The surrogate model was applied recursively given a single reference simulation configuration from MCS 2000 for 100 recursive evaluations. (A) Dice score of the surrogate prediction and initial configuration compared with the simulation state at a given timestep is shown for 100 surrogate model evaluations. The first 10 evaluations of the same data are plotted as picture in picture for clarity. (B) Mean square error (MSE) of the diffusive field concentration plotted for 100 evaluations, the first 10 evaluations of the same data is also plotted as picture-in-picture for clarity. (C) Earth Mover's Distance (EMD) of the distribution of lacunae area for the model prediction and reference configuration are shown for 100 recursive iterations, the first 10 of which are shown as picture-in-picture. Data for all subplots represent mean +/- s.d. of metrics of comparison for predictions and initial reference compared to the true simulation configuration at the stated timestep. Data represents comparison from 25 unique Cellular-Potts model simulations generated for performance evaluation.

(s.d. 9.72, n=25) for the surrogate prediction compared with ground truth, and 21.09 (s.d. 7.77, n=25) for the reference configuration compared with ground truth. These values are plotted as the initial value in the subplots displayed in Fig 3. The surrogate model demonstrated the greatest Dice score on a single iteration of evaluation. Subsequent prediction steps dramatically decrease in Dice score, increase in MSE of the predicted diffusive field, and increase in EMD of the lacunae areas of the predicted configuration. Together, these represent increasingly poor prediction at subsequent simulation steps and divergence in predicted simulation configuration as compared to the Cellular-Potts model configuration. However, the surrogate model maintains greater Dice scores than the reference configuration for 10 recursive iterations, predicting up to 1000 MCS ahead of the given input (Fig 3A), and similar EMD values as the reference configuration for 3 recursive evaluation steps (Fig 3C) demonstrating predictive ability with recursive evaluation for short timescales up to 300 MCS ahead of reference, or 3 recursive iterations. The surrogate model maintains a lower diffusive field MSE than the reference configuration throughout the course of the simulation.

Upon visual inspection of a representative time series of the simulated vessel network configuration (S1 Movie), the surrogate produced fewer sprouts than it should, and the sprouts did not extend as quickly compared to ground truth. Our loss function of combined Dice + MSE (see Methods) is not sensitive to these details. Thus, failing to accurately reproduce these specific behaviors was not heavily penalized during training, making it difficult for the surrogate model to learn these dynamics. Observing chemical field values over a representative simulation time course (S1 Movie), the surrogate appears to predict lower diffusive field concentrations overall.

To further investigate how the surrogate model was failing to reproduce simulation configurations at long timescales, we quantified the number of Cellular-Potts lattice sites corresponding to the vessel network as well as the sum of all diffusive field values at all lattice sites. The surrogate model was unable to maintain both vessel area (Fig 4A) and diffusive field values (Fig 4B) over recurrent evaluations. This represents the mechanism by which the surrogate model diverges from the ground truth and results in poor performance over multiple evaluations.

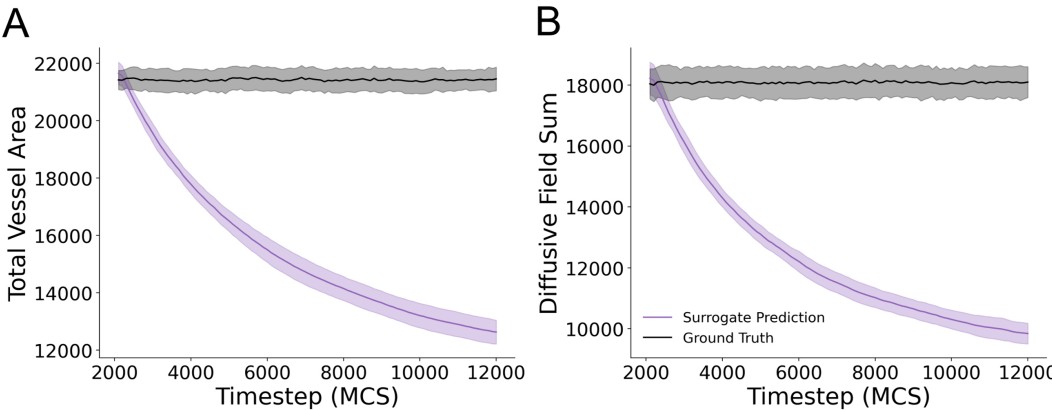

**Fig 4**. **Surrogate model fails to retain vessel area and diffusive field concentrations over multiple recursive evaluations.** (A) Vessel area for predicted model configurations and ground truth simulation configurations for each evaluated timestep for 100 recursive evaluations. Mean and standard deviation are shown for data from 25 unique simulations from the evaluation dataset (B) Sum of the diffusive field concentration at all lattice sites in the predicted and ground truth simulation states. Mean and standard deviation are shown for data from 25 simulations from the evaluation dataset.

## Model computational time comparison

The surrogate model is most accurate at a single predictive evaluation step to predict 100 MCS ahead of an input configuration. To investigate the potential for the surrogate to accelerate evaluation of the Cellular-Potts model, we compared the time required to calculate 100 MCS ahead of a given reference configuration using the trained surrogate versus the native CompuCell3D (CC3D) code. We generated 100 unique model configurations and calculated the amount of time required for the CC3D code to calculate 100 MCS for each configuration. We compared the time required for native CC3D code execution with the time required to evaluate 100 MCS with the model surrogate. The results of this experiment are displayed in Fig 5. CPU/GPU CPU/CPU comparisons are demonstrated.

The trained surrogate model significantly accelerates the evaluation of 100 MCS of the Cellular-Potts model when compared to native CompuCell3D code execution. Median and standard deviations for all configurations are listed in Table 1.

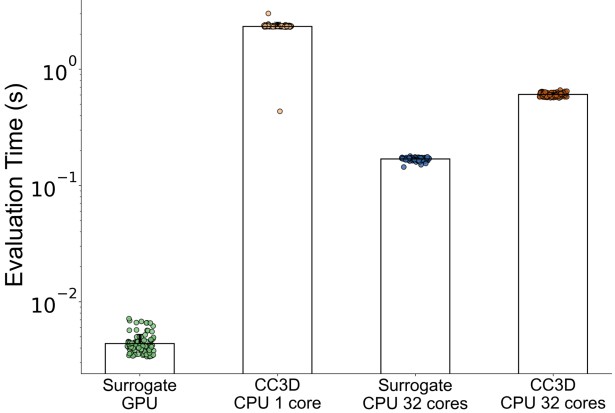

**Fig 5**. **Surrogate model evaluates faster than the original Cellular-Potts model in both CPU/CPU and CPU/GPU comparisons.** Evaluation methods were performed for n=100 independent simulation configurations each. Median surrogate model evaluation on GPU were 0.0041s (s.d. 0.0088) compared with 2.35s (s.d. 0.20) resulting in a speed increase by up to a factor of 562x when comparing median evaluation time. Surrogate model evaluations were performed on an NVIDIA RTX 3090 GPU, CC3D simulation execution was performed on an AMD Ryzen 9 5950X processor.

**Table 1. Time comparison of surrogate model evaluation compared with native CC3D code execution.**

| Evaluation | Median time (s) | Std. time (s) |
|---|---|---|
| Surrogate GPU | 0.004171 | 0.000877 |
| CC3D 1 core | 2.345840 | 0.203766 |
| Surrogate 32 CPU cores | 0.169648 | 0.004992 |
| CC3D 32 CPU cores | 0.607681 | 0.021977 |

Evaluations comparing the surrogate on a GPU with native CC3D code execution utilizing 1 core resulted in an increase in 100 MCS calculation speed by up to a factor of 562. All evaluation runs were performed on the same consumer-grade workstation computer. The surrogate model evaluations were performed on an NVIDIA RTX 3090 GPU, whereas the CC3D code was executed on an AMD Ryzen 9 5950X processor. 1-core and 32-core runs are included as single-core configurations are often advantageous for parallelization on HPC systems when performing parameter studies during Cellular-Potts model development. 32-core runs represent full utilization of the 5950x 16-core, 32-thread processor. Significant differences in the mean of these groups were confirmed by one-way ANOVA ($p<10^{-6}$). All pairwise comparisons were also significant following Tukey's HSD post-hoc analysis ($p<10^{-6}$)

## Discussion

In this work, we approach the prediction of a future simulation state of a Cellular-Potts agent-based model as a segmentation task and apply a configuration of the U-Net neural network architecture to successfully predict Cellular-Potts model configurations 100 MCS in advance of a given input configuration. This demonstrates the potential for convolutional neural networks to perform as a surrogate model for Cellular-Potts models. The surrogate replicates Cellular-Potts model behaviors that are not explicitly encoded in the original model and arise as a result of the interaction of model agents and a PDE-based diffusive field.

The CPM dynamics of vessel sprouting, branch extension and anastamosis, and lacunae contraction occur consistently enough at the 100 MCS timescale that they may be considered quasi-deterministic. Our work demonstrates that these consistent, quasi-deterministic patterns are feasible for a deterministic U-Net architecture to learn when structured as a segmentation task, even when applied to a stochastic biological system. We would therefore expect this segmentation model approach to a CPM surrogate to yield similar success for classes of models of systems that may also be studied using other deterministic mathematical methods. For example, multiple notable PDE-based models have been developed to study *in vitro* vessel patterning such as the seminal work by Gamba and Serini [25–27]. The investigation of this same stochastic biological process lead to the development of the CPM studied in this work.

In addition to replicating Cellular-Potts model dynamics, the surrogate allows for an accelerated time to generate the future simulation state 100 MCS ahead by a factor of 562 when comparing GPU surrogate evaluation with single-core CC3D code execution. This speed advantage is possible because the surrogate model can leverage graphics processing unit (GPU) hardware on a workstation computer to calculate future simulation states and is trained to calculate the future configuration in one evaluation step. The Cellular-Potts models as implemented in the widely used open-source frameworks CompuCell3D [28], Morpheus [29], Chaste [30], and Artistoo [31] to name a few, perform evaluation of the algorithm on the central processing unit (CPU) and explicitly evaluates each MCS, thus resulting in a greater computational requirement and a greater evaluation time. Recently, GPU-accelerated implementations of the Cellular-Potts method have been developed allowing for significant decreases in the computational time required for model evaluation [32,33]. However, this appears to be an area of active research, and the GPU-accelerated Cellular-Potts algorithm implementations are not compatible with the existing widely accessible frameworks that allow for multi-method or multi-scale simulation features such as specifying agent behaviors in response to secreted diffusive chemical fields.

Our surrogate model demonstrated the greatest prediction accuracy during a single prediction step of 100 MCS and exhibited consistently reduced prediction accuracy over multiple evaluations. We demonstrated that this behavior is due to failure to maintain vessel network area and diffusive field concentrations in the simulation space over multiple timesteps as compared to ground truth. Our loss function did not penalize this behavior and future work to define a more optimal training strategy may yield a surrogate model that is more accurate over additional recursive evaluations. Several additional factors likely contribute to the surrogate model's poor performance when applied recursively. The Cellular Potts Model is inherently stochastic, and these stochastic fluctuations affect model configuration during the simulation, causing the model configuration to diverge from the reference configuration in ways that are difficult for the U-Net to predict at long time scales. Convolutional neural networks such as the U-Net presented here are deterministic models and therefore lack the ability to replicate the stochastic nature of the CPM simulation. Additionally, the mathematical convolution and pooling operations that make up the neural network model create a smoothing effect and filter out the noisy cell boundaries present in the original simulation. This may be problematic because the irregular cell shapes and stochastic cell movements in the CPM contribute to the emergent behavior of the model system. Smoothing an image results in prediction error, and when that smoothed image is used as the reference for subsequent prediction steps, the compounded error results in further deviation from the ground truth, resulting in the observed decrease in prediction performance at multiple prediction steps. Deep generative modeling approaches such as variational autoencoders, normalizing flows, or generative denoising diffusion models may be better suited to capture the stochastic behaviors of the CPM methodology [34]. Future efforts may explore these approaches in the context of developing a CPM model surrogate. In ongoing work, our group has demonstrated success at leveraging denoising diffusion models as surrogates for the ABM used here at long timescales of 20,000 MCS for multiple parameter sets [35]. However, our denoising diffusion models demonstrate speed increases that are far less dramatic than those demonstrated by the model described in this work.

The surrogate model is therefore best suited for making single prediction steps but retains predictive ability up to 3 predictive steps (300 MCS) when applied recursively. Considering this, as well as the significant increase in evaluation speed compared to native model execution per evaluation step suggests that the surrogate could be valuable if used alongside the mechanistic CPM model. Future work could explore hybrid approaches, such as alternating between the surrogate and the mechanistic model. The surrogate could be applied for a single prediction step and passed back and forth to a Cellular-Potts solver to incorporate the Potts model stochasticity and reset the reference configuration for the surrogate configuration. While passing data between CPU and GPU could be explored to integrate the surrogate with the CPM, our approach requires converting Potts model data into two-channel images, which may introduce computational overhead.

Our current work currently demonstrates the feasibility of the U-Net as a CPM surrogate for 2-dimensional CPM simulations. 3D simulations using the CPM are also common but are cited as computationally expensive [32]. 3D convolutional neural networks and U-Nets have been documented as successful approaches to 3D segmentation tasks [36] and thus may be extended to serve as surrogates for 3D CPM simulations in future work.

## Conclusion

In summary, we demonstrate the applicability of convolutional neural networks classically applied to segmentation tasks to perform as surrogates of Cellular-Potts agent-based models inclusive of PDE-based diffusive fields. Our surrogate predicts the simulation configuration significantly faster than native Cellular-Potts algorithm execution and can replicate several multicellular emergent tissue-scale behaviors not explicitly encoded in the behaviors of individual CPM agents. While the model is effective at shorter timescales (up to 3 recurrent evaluations), its predictions diverge from the mechanistic CPM simulation over longer timescales. Overall, this work highlights the potential of deep learning approaches to accelerate the evaluation of Cellular-Potts models, offering a promising direction for improving computational efficiency for complex biological simulations.

## Methods

### Cellular-Potts agent-based mechanistic model

We selected the previously published model of vasculogenesis by Merks et al. [1] because it was an adequate example of an agent-based model replicating a biological system implemented using the Cellular-Potts modeling method. The Cellular-Potts (Glazier-Graner-Hogeweg) agent-based model was re-implemented in the CompuCell3D (CC3D) [28] open-source simulation environment version 4.6.0. In the CPM method, individual cells are represented as a collection of pixels on a square, two-dimensional lattice 256x256 pixels in dimension. Cells are given properties of predefined volume, contact energy with surrounding cells and medium, and a tendency to chemotax toward a diffusive cytokine gradient. These properties are defined mathematically using an effective energy functional $H$ shown in Eq 1 below. This effective energy functional is evaluated on a cell-by-cell basis each computational timestep, denoted Monte-Carlo step (MCS) in the CPM method.

$$
\begin{aligned}
H = & \sum_{i,j,\text{neighbors}} J_{\tau(\sigma_i),\tau(\sigma_j)}(1 - \delta_{\sigma_i,\sigma_j}) \\
& + \lambda_{\text{volume}}(V_{\text{cell}} - V_{\text{target}})^2 \\
& + \lambda_{\text{surface}}(S_{\text{cell}} - S_{\text{target}})^2 \\
& + \sum_{i,j} -\lambda_{\text{chemotaxis}} \left[ \frac{c(\mathbf{x}_{\text{destination}})}{sc(\mathbf{x}_{\text{destination}}) + 1} - \frac{c(\mathbf{x}_{\text{source}})}{sc(\mathbf{x}_{\text{source}}) + 1} \right].
\end{aligned}
\tag{1}
$$

Where the first term describes cell contact energy for neighboring cells with a contact coefficient $J$ where $i,j$, describe neighboring lattice sites, $\sigma_i$ and $\sigma_j$ describe individual model agents occupying site $i$ and $j$ respectively, and $\tau(\sigma)$ denotes the type of cell $\sigma$ in the model. The second term defines a volume constraint $\lambda_{\text{volume}}$ applied to each cell where $V_{\text{cell}}$ represents the current volume of a cell at a given point in the simulation, and $V_{\text{target}}$ is the volume assigned to that cell with stiffness $\lambda_{\text{volume}}$. Similarly, the third term applies a surface area constraint, where $\lambda_{\text{surface}}$ determines the influence of assigned circumference $S_{\text{target}}$. The fourth term defines chemotactic agent motility in response to a diffusive gradient where $\lambda_{\text{chemotaxis}}$ is a constraint or influence of the differences in chemical concentration $c(\mathbf{x}_{\text{destination}})$ and $c(\mathbf{x}_{\text{source}})$ of pixel-copy-destination and pixel-copy-source locations of a cell agent each MCS, and $s$ denotes a saturation constant.

At each MCS or computational timestep, the CPM model creates cell movement by selecting random pairs of neighboring voxels $(y, y')$ and evaluating whether one voxel located at $y$ may copy itself to its neighboring pair at $y'$. This voxel copy attempt, denoted $\sigma(y, t) \rightarrow \sigma(y', t)$, occurs with the probability defined by a Boltzmann acceptance function Eq 2 of the change in the effective energy of the system $\Delta H$ previously defined in Eq 1.

$$
\Pr(\sigma(y, t) \rightarrow \sigma(y', t)) = e^{-\max\left(0, \frac{\Delta H}{H'}\right)}
\tag{2}
$$

Diffusive chemical concentration values $c$ at each lattice site are described by Eq 3 below:

$$
\frac{\partial c}{\partial t} = D\nabla^2 c - kc + \text{secretion}
\tag{3}
$$

where $k$ is the decay constant of the diffusive field concentration $c$, and $D$ is the diffusion constant. The secretion term accounts for diffusive field concentration added at lattice sites associated with the location of a CC3D model agent to model secretion of a cytokine by biological cells. Periodic boundary conditions were applied to the simulation domain for both cell positions subject to the Potts algorithm and diffusive field concentrations described by Eq 3.

CPM model parameters were defined to produce visually apparent extension of branches, sprouting of new branches, and shrinking of circular lacunae within the parameter space explored previously by Merks et al. [1]. CPM model parameters values are displayed in Table 2.

## Cellular-Potts model simulation and training data generation

Approximately 1000 CPM cell agents were placed randomly throughout the 256x256 simulation domain at the beginning of the simulation and settled into vascular-like structures over the first 200 Monte-Carlo steps (MCS) of a simulation. Training data for the surrogate model was created by saving model configurations from each MCS of a simulation between 200 and 20,000 MCS. A total of 20 unique simulations were generated to produce training data for the surrogate model. Input and ground-truth pairs for training were created by pairing model configurations of a saved MCS as input with the corresponding configuration 100 MCS ahead in the same simulation. Model configurations consisted of a 2-channel 256x256 input image. The first channel consisted of a binary segmentation of the cell positions of the Cellular-Potts simulation, and the second channel consisted of chemical field concentrations of simulation domain lattice sites. Thus, each simulation yielded 19,700 input, ground-truth pairs consisting of a given simulation state and its corresponding state 100 timesteps ahead in the same simulation.

## Surrogate model architecture, training, and performance evaluation

A U-Net architecture was specified with circular padding in convolutional blocks to account for the periodic boundary conditions present in the Cellular-Potts model, and parametric rectified linear unit (PReLU) activations. A demonstrative schematic of the model architecture is displayed in Fig 6. We organized our generated training data into an 80%-20% train-test split for U-Net surrogate model training. We trained the U-Net architecture on the generated input, ground-truth pairs for 100 epochs using a combined loss of binary cross-entropy for the cell-segmentation layer and mean-squared-error (MSE) for the chemical field layer. The influence of the prediction of the two image layers on the loss function was balanced by multiplying the MSE loss by 10.

Surrogate model performance was evaluated on a separate dataset consisting of input, ground-truth pairs from 20 new simulations not included in the set used for training. Model performance was evaluated using dice score as a metric for the cell-segmentation layer, and MSE as a metric for the chemical field layer. The distance in the distribution of lacunae areas between simulated configuration and ground truth as quantified by Earth Mover's Distance (EMD) was also used as a metric to evaluate trained model performance (See Results).

We defined a process to calculate distributions of lacunae areas for a given simulation so that the distributions could be compared. We first augmented the image by patterning it above, below, to the left, and to the right of the original configuration to account for the periodic boundary conditions present in the original Cellular-Potts model. We then labeled all

**Table 2. CPM parameter table.**

| Parameter | Parameter Description | Parameter Value |
|---|---|---|
| $\lambda_{volume}$ | Influence of volume constraint | 5 |
| $V_{target}$ | Number of voxels per cell | 50 |
| $\lambda_{surface}$ | Influence of surface constraint | 1 |
| $S_{target}$ | Number of voxels in cell circumference | 16.8 |
| $J_{cell,medium}$ | Contact energy between cell-medium interface | 8.2 |
| $J_{cell,cell}$ | Contact energy between cell-cell interface | 6 |
| $\lambda_{chemotaxis}$ | Influence of chemotaxis | 2000 |
| $s$ | Saturation constant for chemotaxis | 0.5 |
| $k$ | Decay constant for chemical field | 0.6 |
| $H'$ | Temperature for the Cellular-Potts algorithm | 8 |

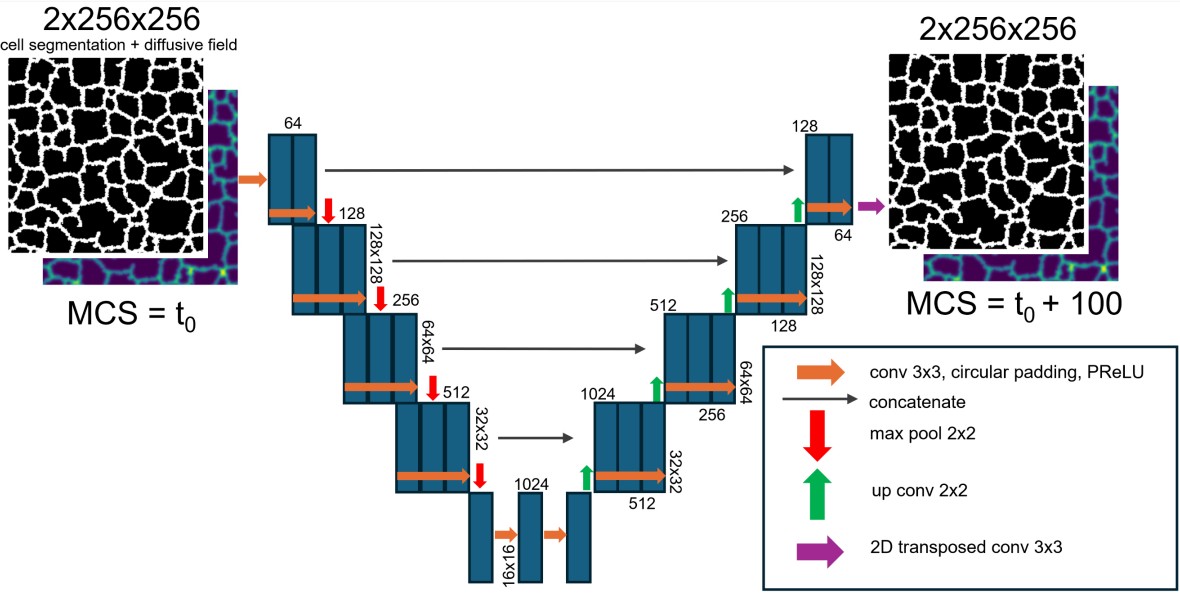

**Fig 6**. **Surrogate model architecture.** U-Net Neural Network model configuration is illustrated. Neural network layers and activations are described to allow for prediction of a mechanistic model simulation configuration 100 MCS in advance of an input configuration.

connected lacunae regions and dropped all duplicate lacunae areas. Duplicate lacunae areas were removed by calculating image inertia tensor eigenvalues for the list of all identified lacunae. This operation defines a mathematical representation of lacunae shape from which we use to ignore duplicate regions. Isolated image regions representing lacunae were labeled and inertia tensors for these regions were calculated using the open-source image processing library scikit-image [37]. We further filtered the set of unique domains by removing all regions with an area less than 3 lattice sites, as vascular lacunae in the simulation are generally large and 1-3 lattice regions rarely appeared due to the stochastic lattice-site exchanges during the Cellular-Potts algorithm. This yielded a list of areas of non-duplicate lacunae for a given simulation state that accounts for the periodic boundary conditions of the Cellular-Potts model. The distribution of areas contained in this list was used to calculate and compare distribution distances between predicted and ground-truth simulation states.

## Supporting information

**S1 Movie. Surrogate model vascular network configuration prediction with recurrent evaluation.** Ground truth (left) from the Cellular-Potts model is compared with the surrogate model prediction (right) given the same initial configuration. (MP4)

**S2 Movie. Surrogate model diffusive field prediction with recurrent evaluation.** Ground truth (left) from the Cellular-Potts model is compared with the surrogate model prediction (right) given the same initial configuration. (MP4)

## Data availability

Code to generate data and specify models used in this work is publicly available at https://github.com/tc2fh/CPM_UNet_Surrogate. This code is also published on Zenodo at https://zenodo.org/records/15399533 [38] with a Creative Commons Attribution 4.0 International license.

## Author contributions

**Conceptualization:** Tien Comlekoglu, J. Quetzalcóatl Toledo-Marín, Geoffrey Fox, James A. Glazier.

**Data curation:** Tien Comlekoglu, J. Quetzalcóatl Toledo-Marín, Tina Comlekoglu, James A. Glazier.

**Formal analysis:** Tien Comlekoglu, J. Quetzalcóatl Toledo-Marín, Tina Comlekoglu, Geoffrey Fox, James A. Glazier.

**Funding acquisition:** Douglas W. DeSimone, Shayn M. Peirce, Geoffrey Fox, James A. Glazier.

**Investigation:** Tien Comlekoglu, J. Quetzalcóatl Toledo-Marín, Geoffrey Fox, James A. Glazier.

**Methodology:** Tien Comlekoglu, J. Quetzalcóatl Toledo-Marín, Tina Comlekoglu, Geoffrey Fox, James A. Glazier.

**Project administration:** Tien Comlekoglu, J. Quetzalcóatl Toledo-Marín, Douglas W. DeSimone, Shayn M. Peirce, Geoffrey Fox, James A. Glazier.

**Resources:** Tien Comlekoglu, J. Quetzalcóatl Toledo-Marín, Tina Comlekoglu, Douglas W. DeSimone, Shayn M. Peirce, Geoffrey Fox, James A. Glazier.

**Software:** Tien Comlekoglu, J. Quetzalcóatl Toledo-Marín, Tina Comlekoglu, James A. Glazier.

**Supervision:** Tien Comlekoglu, J. Quetzalcóatl Toledo-Marín, Douglas W. DeSimone, Shayn M. Peirce, Geoffrey Fox, James A. Glazier.

**Validation:** Tien Comlekoglu, J. Quetzalcóatl Toledo-Marín, Tina Comlekoglu, Geoffrey Fox.

**Visualization:** Tien Comlekoglu, J. Quetzalcóatl Toledo-Marín, Tina Comlekoglu.

**Writing – original draft:** Tien Comlekoglu, J. Quetzalcóatl Toledo-Marín, Tina Comlekoglu, Geoffrey Fox, James A. Glazier.

**Writing – review & editing:** Tien Comlekoglu, J. Quetzalcóatl Toledo-Marín, Tina Comlekoglu, Geoffrey Fox, James A. Glazier.

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
