## [Decision Letter · Decision Letter 0]

8 Jul 2025

PCOMPBIOL-D-25-00953

Surrogate modeling of Cellular-Potts agent-based models as a segmentation task using the U-Net neural network architecture

PLOS Computational Biology

Dear Dr. Glazier,

Thank you for submitting your manuscript to PLOS Computational Biology. After careful consideration, we feel that it has merit but does not fully meet PLOS Computational Biology's publication criteria as it currently stands. Therefore, we invite you to submit a revised version of the manuscript that addresses the points raised during the review process.

Please submit your revised manuscript within 60 days Sep 07 2025 11:59PM. If you will need more time than this to complete your revisions, please reply to this message or contact the journal office at ploscompbiol@plos.org. Please include the following items when submitting your revised manuscript:

We look forward to receiving your revised manuscript.

Kind regards,

Philip K Maini

Academic Editor

PLOS Computational Biology

Mark Alber

Section Editor

PLOS Computational Biology

**Journal Requirements:**

2) Please ensure that the funders and grant numbers match between the Financial Disclosure field and the Funding Information tab in your submission form. Note that the funders must be provided in the same order in both places as well. Currently, the Financial Disclosure states there was no funding received.

**Reviewers' comments:**

Reviewer's Responses to Questions

**Comments to the Authors:**

Reviewer #1: This submission explores how machine learning can be used to significantly accelerate computationally expensive mechanistic models, focusing on the Cellular Potts Model (CPM). Given the CPM’s popularity and broad application, the work addresses an important and timely question with potential implications for simulating larger and more complex biological systems.

The manuscript is well-written and easy to follow. The authors clearly describe the benefits of using a machine learning-based surrogate model, including improved computational speed and the ability to reproduce key features of the CPM such as vessel sprouting, branching behavior, and lacunae contraction. They also acknowledge the model’s limitations, notably its inability to prevent gradual lacunae growth—an effect not present in the original model.

The figures are generally clear and informative. However, the numbering of figures in the main text does not align with the numbering in the figure captions, which should be corrected for consistency and ease of reference.

Main Concerns

My principal concern is that the success of the surrogate model is overstated in several key parts of the paper. While the authors acknowledge in the main text that the surrogate performs well for only one to three prediction steps, they still present the model as broadly useful in ways that are not fully supported by the results. This limitation substantially reduces the model’s practical utility, as the performance gain from faster simulations cannot be fully leveraged over such short prediction horizons.

For example, the abstract states:

“The surrogate model was trained to predict 100 computational steps ahead (Monte-Carlo steps, MCS), accelerating simulation evaluations by a factor of 590 times compared to CPM code execution. Over multiple recursive evaluations, our model effectively captures the emergent behaviors demonstrated by the original Cellular-Potts model such as vessel sprouting, extension and anastomosis, and contraction of vascular lacunae.”

While technically accurate, this statement omits the critical issue that the model's predictions deteriorate over longer time horizons, particularly in representing lacunae size. For balance, the abstract should briefly mention these limitations.

Similarly, at the end of the Introduction, the authors claim:

“Our approach demonstrates the potential for deep neural networks to serve as efficient surrogate models for CPM simulations, enabling faster evaluation of computationally expensive CPM of biological processes at greater spatial and temporal scales.”

Given the current model’s performance, this statement is overly optimistic. While the long-term potential of such approaches is promising, this particular implementation does not yet fulfill the goals set forth in the statement.

In the Conclusion, the authors write:

“Our surrogate predicts the simulation configuration significantly faster than native Cellular-Potts algorithm execution and can replicate emergent behaviors present but not explicitly defined in the Cellular-Potts model.”

Again, this claim should be tempered by noting that the speed-up is only relevant for very short simulation times, beyond which the surrogate model’s predictions become unreliable.

Minor Comment

• The p-value reported in Figure 5 (1 × 10⁻¹⁸³) is unnecessary. Given the visual clarity of the differences between models, such an extreme level of significance adds little value to the reader’s understanding.

Reviewer #2: This paper presents a U-Net CNN as a surrogate model to predict future

behaviour of a two-dimensional individual cell cellular Potts model.

1. The study seems preliminary and inconclusive. While certainly

worthwhile of publication, it would be preferred if the usefulness of

predicting a mere 100 MCS could be shown to be useful. For example,

if the hypothesised scheme presented in the paragraph at line 234

could be shown to give quantifiable, useful results in half the time of

a cellular Potts model, then the paper would have a positive story to

tell.

2. While it's reasonable to dismiss predicting over short/long time spans

it is completely unclear why 100 MCS was selected as the quantity for

prediction. Surely this is a hyper-parameter which ought to be study

and tuned as part of the model?

3. The reading experience was impaired by presenting the figures in a

permuted order. (The U-net architecture figure referred to as Figure 6

in the Supplementary appeared as Figure 1.)

4. Anastomosis and lacunae present different prediction problems in 3D

where cellular Potts models become unfeasibly slow. Is it possible

that the surrogate could be even more useful in 3D?

5. Ref 32 mentioned on line 32 seems to be key to whether the

current study is useful or a poor substitute. What is the publication

status of Ref 32?

Minor points

------------

l11 "of as vessel"

l235 "3 predictive steps" -> "3 predictive steps (300 MCS)" (for

clarity, if that is what you mean).

l256 Hamiltonian equation appears out of nowhere. Introduce it.

l346 Fix double quotes

l173 Is it possible to show CPU/CPU or GPU/GPU time comparisons?

Ref 8: "potts" -> "Potts"

Ref 28: "Elife" -> eLife"

Reviewer #3: This paper presents an approach for surrogate modeling for the cellular Potts modeling (CPM) framework, which is evaluated on an example model of angiogenesis that involves both a diffusive field and a grid with cells.

Surrogate modeling is an interesting approach that is being applied successfully to many complex simulation methods, e.g., molecular dynamics, and so it is interesting and relevant to investigate similar speedups for the CPM. Unfortunately, I think that the results presented in this paper are still too preliminary to be broadly relevant, and I consider it unlikely that the method proposed in this paper is actually useful; it is currently more an interesting proof of concept for people working in this field, but fundamentally different methods will probably need to be used to actually do surrogate modeling for the CPM.

The main issue is simply that the CPM is a stochastic model, and so the premise of predicting the next configuration after say 100 steps is to some extent flawed. If we ran the CPM many times with different random seeds, it would also not produce the exact same configuration. However, in this paper this task is treated as a "segmentation problem" which is fundamentally a deterministic problem. Using a deterministic surrogate for a stochastic model would have profound implications that may make it very hard or even impossible to use the surrogate for the same research questions as the original model. There may be possible ways around this; for example there could be a subclass of models where the randomness itself may not be critical and simulating a "typical" trajectory may be enough; however without any discussion of such issues or examples how this kind of surrogate could be used the paper is unfortunately too preliminary to be useful.

The authors are well aware of this issue, as they mention it in their discussion section, and have actually already begun a separate project where they use a denoising diffusion model as an architecture for the surrogate, which seems like a much better approach (https://arxiv.org/abs/2505.09630).

In summary, this manuscript would be much stronger if it showed that there is an advantage of this deterministic approach compared to the seemingly more suitable one of using a stochastic generative model as a surrogate. I cannot currently discern such an advantage from the data being shown. While the results do demonstrate that the deterministic surrogate makes predictions that retain some characteristics of the actual model, they do not demonstrate that we can still generate the same kind of insights from it.

**Have the authors made all data and (if applicable) computational code underlying the findings in their manuscript fully available?**

Reviewer #1: Yes

Reviewer #2: Yes

Reviewer #3: Yes

PLOS authors have the option to publish the peer review history of their article (what does this mean?). If published, this will include your full peer review and any attached files.

Reviewer #1: No

Reviewer #2: No

Reviewer #3: No

**Figure resubmission:**
---

## [Decision Letter · Decision Letter 1]

16 Oct 2025

Dear Prof. Glazier,

We are pleased to inform you that your manuscript 'Surrogate modeling of Cellular-Potts agent-based models as a segmentation task using the U-Net neural network architecture' has been provisionally accepted for publication in PLOS Computational Biology.

Best regards,

Philip K Maini

Academic Editor

PLOS Computational Biology

Mark Alber

Section Editor

PLOS Computational Biology

Reviewer's Responses to Questions

**Comments to the Authors:**

Reviewer #1: My concerns and suggestions have been incorporated into the text and I have no more concerns.

Reviewer #2: Authors have addressed my criticisms (and those of another reviewer) to my satisfaction.

Reviewer #3: Thank you for your responses to my concerns. I see that you have added a new paragraph to the discussion emphasizing again why you feel your approach can be useful. While I would have liked to see an explicit acknowledgement of the significant limitations that the current proof of concept still has, I have no further comments or requests at this stage.

**Have the authors made all data and (if applicable) computational code underlying the findings in their manuscript fully available?**

Reviewer #1: Yes

Reviewer #2: Yes

Reviewer #3: **No: **

PLOS authors have the option to publish the peer review history of their article (what does this mean?). If published, this will include your full peer review and any attached files.

Reviewer #1: No

Reviewer #2: No

Reviewer #3: No

---

## [Editor Report · Acceptance letter]

PCOMPBIOL-D-25-00953R1

Surrogate modeling of Cellular-Potts agent-based models as a segmentation task using the U-Net neural network architecture

Dear Dr Glazier,

I am pleased to inform you that your manuscript has been formally accepted for publication in PLOS Computational Biology. Your manuscript is now with our production department and you will be notified of the publication date in due course.

With kind regards,

Anita Estes
